# Surface Water Quality Differs between Functionally Similar Restored and Natural Wetlands of the Saint Lawrence River Valley in New York

Brendan Carberry, Tom A. Langen 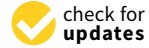 and Michael R. Twiss *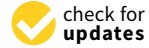

Department of Biology, Clarkson University, Potsdam, New York, NY 13699, USA;
brendanjcarberry@gmail.com (B.C.); tlangen@clarkson.edu (T.A.L.)
* Correspondence: mtwiss@clarkson.edu; Tel.: +1-315-268-2359

**Abstract:** We tested the hypothesis that upland wetland restorations provide the same quality of wetland, in terms of ecosystem services and biodiversity, as natural wetlands in the St. Lawrence River Valley. Water quality (pH, alkalinity, colored dissolved organic matter, phytoplankton community composition, chlorophyll-a, fecal coliform, total phosphorus, dissolved nitrate, turbidity, specific conductivity) in 17 natural and 45 restored wetlands was compared to determine whether wetland restoration provided similar physicochemical conditions as natural wetlands in the Saint Lawrence River Valley of northeastern New York State. Natural wetlands were more acidic, which was hypothesized to result from the avoidance of naturally acidic regions by farmers seeking to drain wetlands for crop and pasture use. Natural wetlands had significantly greater fecal coliform concentrations. Restored wetlands had significantly greater specific conductivity and related ions, and this is attributed to the creation of wetlands upon marine clay deposits. Other water quality indicators did not differ between restored and natural wetlands. These findings confirm other research at these same wetlands showing no substantial differences between restored and natural wetlands in major biotic indicators. Thus, we conclude that wetland restoration does result in wetlands that are functionally the same as the natural wetlands they were designed to replicate.

**Keywords:** cyanobacteria; phosphorus; restoration ecology; water quality; wetland



## 1. Introduction

Wetland restoration is a form of ecological engineering wherein these valued ecosystems in the landscape are reestablished for communal (human, ecosystem) good [1]. Public-private partnerships for wetland restoration are a mechanism by which private landowners and government agencies work together to improve the environmental quality of a human-modified landscape. US federal wetland restoration programs such as those administered by the US Department of Agriculture's Natural Resources Conservation Service (NRCS) or US Fish and Wildlife Service (FWS) are intended to restore wetlands and the ecosystem services wetlands provide on agricultural landscapes where wetlands have been drained or degraded in the past [2,3]. Wetlands are important features in the Upper St. Lawrence Valley landscape that provide numerous ecosystem services such as fish and wildlife habitats, natural water quality improvement, flood protection, opportunities for recreation, and aesthetics. NRCS and FWS collaborate with private landowners to restore or enhance wetlands on former or currently productive agricultural lands. In the St. Lawrence River Valley of New York, over 200 landowners have had wetlands restored on their property via these programs. Evaluating the success of wetland restorations is essential to program expansion and the design of the best approach to achieve communal benefits. A key question to evaluate programs success is "Do these restorations provide the same quality of wetland, in terms of ecosystem services and biodiversity, as natural wetlands?"

Regional assessments of wetland restoration programs indicate that aggregating over restoration projects, wetland restoration programs do augment ecosystem services in agricultural landscapes [4,5], though not necessarily the same quality as the former natural wetlands that had been lost on the landscape due to drainage or other hydrological alterations. There is a lack of studies, however, that evaluate restored wetlands on a project level to natural wetlands in the same landscape [6,7].

We conducted biotic surveys and informational surveys of landowners at a large set of restored and similar natural wetlands within the St. Lawrence River Valley of New York. Reference natural wetlands were similar in landscape context and size to wetlands restorations, and in proximity to them. Landowners had voluntarily enrolled in wetland restoration programs because they want to improve the environmental quality of their property by establishing and protecting well-functioning wetlands [8]. Restored wetlands were similar to natural wetlands in terms of birds, amphibians, reptiles, fish, and vegetation [6,8,9]. By various ecological indicator metrics, restored wetlands were qualitatively similar to natural wetlands, albeit quantitatively most indices scores, on average, were a little lower (i.e., lower environmental quality) than natural wetlands [10]. In comparison, these restored wetlands scored much higher (i.e., better environmental quality) that wetlands in a nearby Great Lakes Area of Concern.

One wetland ecosystem service is improvement of water quality, and water quality is used as one indicator of wetland state. Thus, one way to evaluate the success of wetland restoration programs at restoring well-functioning wetlands is to compare water quality between restorations and natural wetlands in the same landscape. We used a modified water quality index developed for coastal marshes in the Laurentian Great Lakes [11]; this index used water quality parameters that are significantly related to Great Lakes basin-wide land use stressors and sensitive to road density [12]. Our implementation of the index incorporated water turbidity, pH, temperature, conductivity, total nitrogen, total phosphate, and chlorophyll-a. We found that the water quality index was, on average, quantitatively slightly lower (indicating poorer water quality) than natural wetlands. However, the water quality index averaged much higher (i.e., better water quality) than wetlands in the nearby Great Lakes Area of Concern at Massena/Akwesasne, where significant anthropogenic stressors are known to be present [10]. However, this water quality index, surprisingly, was not correlated with other biotic and landscape indices of wetland quality. We surmised in [10] that water quality was a poor indicator of wetland habitat quality for wetland-associated plants and animals, and this may be because water quality parameters in shallow wetlands are highly variable at short timescales and within short distances.

Water quality properties were compared between sets of restored and natural wetlands to determine if adverse effects are a result of wetland restorations, e.g., nutrient enrichment of the aquatic environment leading to eutrophication, and the related increase in potentially toxigenic cyanobacteria (commonly referred to as blue-green algae). If adverse effects were detected in water quality, then social impacts could reduce program acceptance and efficiency. Here we examine water quality attributes between a set of restored and natural wetlands in the St. Lawrence River Valley of New York, including chemical, physical, and microbial parameters that are indicators of nutrient runoff-associated eutrophication and other anthropogenic stressors.

## 2. Materials and Methods

Wetland sampling for water quality was conducted on 17 natural wetlands and 45 restored wetlands (Figure 1) over a four-week period (25-July-2014 to 25-August-2014); this set of wetlands underwent extensive biotic assessment in 2009–2011 and 2014 [6]. Wetland restoration techniques included removal of drainage tiles, blocking drainage ditches, excavation of potholes, creation of dikes and berms, and installation of water control structures on outflow streams. Reference natural wetlands were selected to match restorations in terms of size and landscape context (see [6] for details on site selection and geographic dispersion). The wetlands were shallow (under 2 m maximum depth) and

small (1–3 ha surface area), with bordering upland vegetation that varied from old-field to hardwood and coniferous forest. Introduced and invasive wetlands plants were present at most wetlands [9,13]. Landowners rarely managed water levels using the water control structures [8]. Most wetlands with adjacent forest had signs of beaver (*Castor canadensis*) activity, and all likely had muskrat (*Ondatra zibethicus*) present [6].

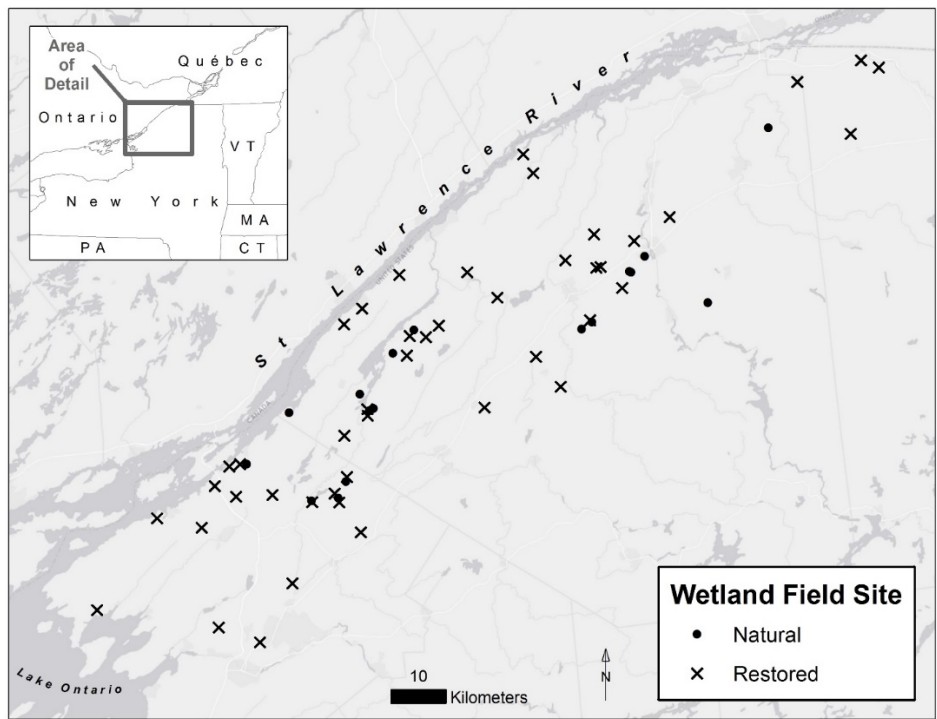

**Figure 1.** Location of sampled natural (*n* = 17) and restored (*n* = 45) wetlands in the Saint Lawrence River Valley, northern New York.

Grab samples (one liter) were collected in acid-clean polycarbonate bottles from the surface water present in each wetland, stored cool in the dark, and processed that day. Wetlands were sampled on a schedule determined by the logistics of travel; sampling was avoided after heavy rainfall by going into the field minimally three days after a thunderstorm in the area.

We measured 12 physicochemical and biological state variables: total chlorophyll-a (acetone extraction and quantification by fluorometry [14]; phytoplankton community (pigment-specific fluorometry); fecal coliform bacteria (Petri-Film; 3M Corp.); dissolved nitrate, sulfate, and chloride by ion exchange chromatography; colored dissolved organic matter (CDOM) by fluorometry (TD-700 using Suwanee River fulvic acid reference material (International Humic Substances Society); turbidity by absorbance at 500 nm in a 5 cm path length cuvette; pH by potentiometry, and alkalinity by Gran titration using HCl; specific conductivity was measured using an electronic meter (YSI model 600XL); and total phosphorus (TP) by colorimetry following persulfate digestion at 121 °C [15]. Dissolved solutes were measured after filtration through a 0.2-μm polyether sulfone membrane syringe filter (Whatman). All measurements were made using standard limnological and analytical methods. Water temperature and dissolved oxygen were not measured due to their inherent high magnitude of diel variation in wetlands.

The phytoplankton community composition was assessed using the FluoroProbe (bbe Moldaenke, GmbH), an instrument capable of classifying the community into four major phytoplankton groupings based on pigment content [16]. Each sample was corrected for background fluorescence using water filtered through 0.2-μm pore-size syringe filters prior to evaluating the non-filtered sample.

Statistical hypothesis tests of specific water quality parameters between restored and natural wetlands were done using Student's *t*-test for unequal variances on non-transformed data, with two-tailed distributions.

## 3. Results and Discussion

No significant difference (*t*-test; $p < 0.05$) were observed between natural and restored wetlands for CDOM, turbidity, phytoplankton community composition, nitrate, total chlorophyll-a and total phosphorus (Figure 2). Significant differences were observed for chloride ($p = 0.010$) and sulfate ($p = 0.014$) concentrations, alkalinity ($p = 0.006$), specific conductivity ($p = 0.002$), pH ($p = 0.001$), and fecal coliform concentrations ($p = 0.005$). The complete data set is an electronic appendix at https://data.mendeley.com/datasets/m7 dycy7gt6/2, accessed on 30 May 2021. Although the analysis in this study is based on a measurement campaign in a single season, we acknowledge that there might be seasonal differences within one waterbody. Repeated measurements of water quality parameters over a summer season in a smaller subset of these wetlands showed that water quality parameters were consistent between sampling dates over five months [13]. Thus, we believe that single visit to this larger set of wetlands over a short duration (one month) was adequate for assessing differences.

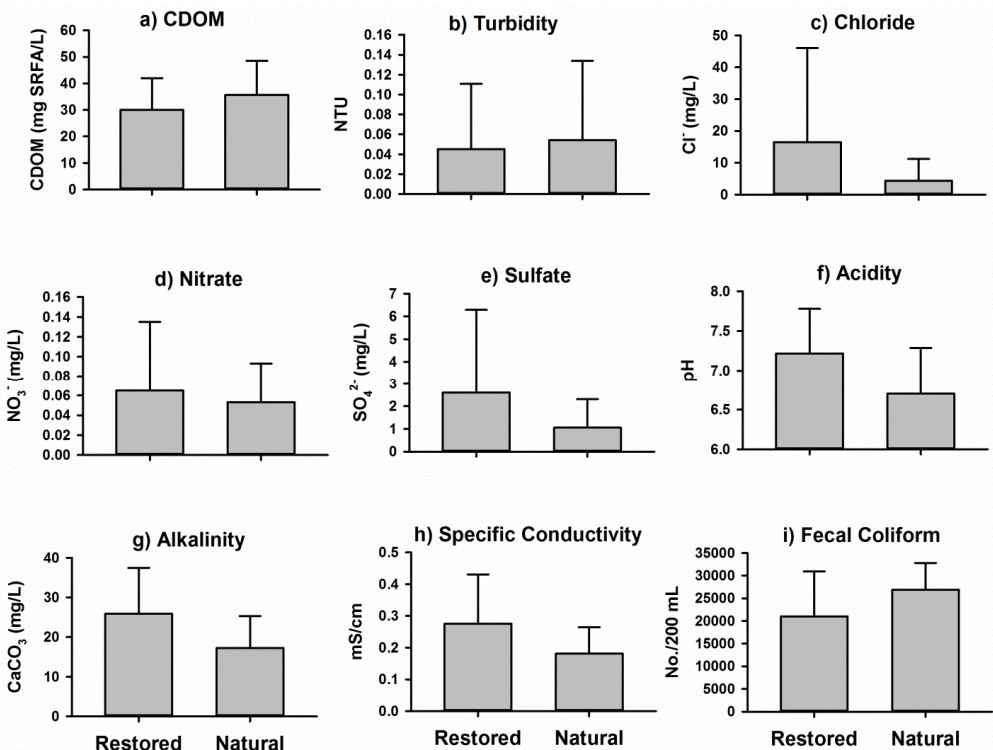

**Figure 2.** Observed water quality in a set of restored (*n* = 45) and natural (*n* = 17) wetlands found in the Saint Lawrence River Valley in northern New York. Significant difference between wetland types ($p < 0.05$) were present for: (**a**) CDOM, (**b**) turbidity, (**c**) chloride, (**d**), nitrate, (**e**) sulfate, (**f**) acidity, (**g**) alkalinity, (**h**) specific conductivity, and (**i**) fecal coliform. Values are mean ± SD.

Total mercury and methylmercury was determined (see [13] for details) in surface waters from four natural and 16 restored wetlands from among the set described here. The wetlands were sampled three to five times at approximately monthly intervals (over the period of May to October 2015). There was no significant difference between mercury concentration and mercury speciation between the two types of wetlands [13]: total mercury and percentage mercury in natural and restored wetlands was 1.0 ± 0.4 ng/L (37 ± 17%) and 1.1 ± 0.5 ng/L (46 ± 15%), respectively (values are mean ± standard deviation; SD).

Both natural and restored wetlands had similar phytoplankton community composition (Table 1). We were most interested to determine if restored or natural wetlands contained more phycocyanin-rich cyanobacteria, a group of phytoplankton that contain species capable of producing potent toxins such as microcystins and anatoxins [17]. There was no significant difference in the proportion of phycocyanin-rich cyanobacteria between the wetland types nor the absolute amount of potentially toxigenic phytoplankton between wetland types (Table 1).

**Table 1.** Phytoplankton community composition observed in vivo in restored (*n* = 45) and natural (*n* = 17) wetlands in the Saint Lawrence River Valley of northern New York, measured using spectrofluorometry. Total chlorophyll-a (Chl-a) was measured following solvent extraction. PC = phycocyanin; PE = phycoerythrin. Values are mean ± SD.

| Wetland Type | Phytoplankton Groups (% Total) | | | | |
|---|---|---|---|---|---|
| | Chlorophyta and Euglenophyta | PC-Rich Cyanobacteria | Pyrrophyta and Heterokontophyta | PE-Rich Cyanobacteria and Cryptophyta | Total Chl-*a* (μg/L) |
| Restored (*n* = 45) | 38 ± 24 | 19 ± 19 | 35 ± 22 | 7.2 ± 13 | 29 ± 50 |
| Natural (*n* = 17) | 36 ± 19 | 24 ± 17 | 30 ± 18 | 10 ± 17 | 18 ± 15 |

Chlorophyll-a content within wetlands was highly variable and both wetland types were highly productive on average, as seen by mean chlorophyll-a at the 20 μg/L threshold for eutrophy (Table 1); this is not surprising given wetlands are shallow aquatic systems and highly productive during summer. There was no significant difference between phytoplankton groupings in natural or restored wetlands, based on pigment-based groupings of the phytoplankton community.

Significant differences (*t*-test; $p < 0.05$) between restored and natural surface water quality parameters were detected for fecal coliform concentrations, pH, alkalinity, and specific conductivity. Natural wetlands had 30% greater fecal coliform concentrations. Natural wetlands (pH 6.70) were 3.3 times more acidic than restored wetlands (pH 7.22), calculated by comparing {H+} derived from lab pH. Natural wetlands had 1.5-times lower alkalinity and specific conductivity than restored wetlands.

Other potential sources of dissolved ions in wetland surface waters were sewage and proximity to roads. Although there are significantly more fecal coliforms in natural wetlands, there is no indication of sewage input from human sources or manure run off from livestock farming, based on visits to these sites. Road salt applications for winter road management can have profound impacts on roadside waterways and groundwater [18]. Reference natural wetlands were more distant to roads than restorations on average (natural: 352 ± SD 305 m, restoration: 152 ± 135 m), but the average wetland was distant enough that it was unlikely that elevated chloride was from deicing road salt. Moreover, for restorations there was no correlation between distance to a road and chloride or conductivity (distance-chloride r = −0.01, one tailed $p = 0.5$, distance-conductivity r = −0.13, $p = 0.2$); the two highest chloride concentrations were at wetlands 100 m from a potential source road.

Although TP was positively associated with specific conductivity in natural wetlands (r = 0.48, one-tailed $p < 0.03$) with a similar trend in restored (r = 0.23, $p = 0.06$) and there was significantly greater fecal coliform densities in natural wetlands (Figure 2), there was no visible evidence during site visits that natural wetlands were impacted by human sewage or manure spreading. Thus, it appears that restored wetlands had greater concentrations of solutes in them; however, this did not affect the trophic status of the restored wetlands, as could be inferred from differences in total phosphorus concentrations ($TP_{Restored}$ = 32 ± 22 μg/L, $TP_{Natural}$ = 32 ± 24 μg/L).

We hypothesize that the significantly greater salt content (as indicated by specific conductivity and chloride) in restored wetlands is due to the creation of these wetlands,

which often occurs by simply scraping off top soil until an impervious clay layer is reached or creating a dike in a region that has impervious soil layers (e.g., clay). Clays in the St. Lawrence River Valley are remnants of glacial activity and the Champlain Sea that existed in this area as late as 6000 years ago when these marine clays deposited under briny conditions [19]. There is a significant difference in the molar ratio ($SO_4^{2-}$:$Cl^-$) between restored ($0.20 \pm 0.32$) and natural ($0.43 \pm 0.54$) wetlands. The molar ratio of sulfate to chloride is exceeded in all wetlands relative to seawater ($SO_4^{2-}$:$Cl^-$ $\approx 0.05$), which suggest that there has been more chloride flux from the clays in to overlying fresh waters. Chloride would be more mobile from clays, and more so with exposed clays as in restored wetlands. In support of these observations, a long-term study of restored wetlands in a comparable wet landscape in central New York concluded that establishment of soil conditions critical for water quality in restored wetlands can require decades to centuries to reach reference conditions [20].

Natural wetlands were more acidic but this was not solely due to higher concentrations dissolved weak organic acids (humic and fulvic acids) as indicated by CDOM concentration (Figure 1). Natural wetlands frequently had mature conifer tree stands growing along the margins; the acidifying effects of conifer litter may have caused the lower pH. There is a history of high and sustained levels of atmospheric sulfate deposition in this region that would contribute to sulfate content, reduction in alkalinity, and increased acidity in the ground and surface water [21], but this would affect restored and natural wetlands alike.

We suspect that wetlands that were in naturally acidic areas had adjacent acidic (and hence low quality) soils and were thus avoided by farmers due to poor soil condition and their collective traditional ecological knowledge of crop production in such soil. Hence, this set of natural wetlands were those selected to remain in a natural state. Further study (sensu [22]) would be required to examine whether soil infertility explains the more acidic natural wetlands in this region.

## 4. Conclusions

We conclude that water quality is similar between natural and restored wetlands in the St Lawrence River Valley, and any differences are minor and may be a result of how the restoration projects were done (site selection for wetland restoration) or the result of site characteristics of remnant natural wetlands (agricultural bias against removing a wetland from the landscape). From a water quality perspective, some differences exist between wetland types (salinity, pH, fecal coliform content) yet they do not have an impact on criteria of interest, such as trophic status (as indicated by the concentration of phosphorus) or the abundance of potentially toxigenic cyanobacteria. Overall, wetland restoration programs do meet their objectives of providing wetlands that are functionally similar to natural wetlands on the landscape.

**Author Contributions:** B.C.: investigation; T.A.L.: funding acquisition, project administration, writing—review and editing; M.R.T.: conceptualization, investigation, writing—original draft, review and editing. All authors have read and agreed to the published version of the manuscript.

**Funding:** This research was funded by a grant from the University of Michigan Water Center and the Erb Family Foundation.

**Data Availability Statement:** The datasets generated and analyzed during the current study are available in the Mendeley Data repository: https://data.mendeley.com/datasets/m7dycy7gt6/2, accessed on 30 May 2021.

**Conflicts of Interest:** The authors declare that they have no known competing financial interests or personal relationships that could have appeared to influence the work reported in this paper.

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
