# Peer review of "Surface Water Quality Differs between Functionally Similar Restored and Natural Wetlands of the Saint Lawrence River Valley in New York"

_land, doi:10.3390/land10070676_

Round 1

Reviewer 1 Report

This manuscript compares the functionality of restored wetlands to that of natural wetlands in the St. Lawrence River valley in New York State.  This work is important to understand if restored wetlands can effectively reestablish the ecosystem services lost when natural wetlands are drained or degraded as a result of human activity. 

The authors hypothesize that restored upland wetlands provide the same ecosystem services and biodiversity as natural wetlands in the St. Lawrence River valley. 

The authors compared ten water quality measures from 17 natural and 45 restored wetlands to determine if wetland restoration results in comparable physiochemical conditions to those found in natural wetlands.  Natural wetlands were found to be more acidic and have higher fecal coliforms.  Restored wetlands had higher conductivity and related ions.  The other water quality parameters measured showed no significant difference, leading the authors to conclude that the restored wetlands function in a similar way to the natural wetlands in the region.

Specific Comments:

Title: It feels like there is a disconnect between the title and the paper.  The conclusion statement used as the title seems to overstate the main conclusion in the paper.  The conclusion section states that water quality differences are minor, while the title suggests large differences in wetlands that have similar functions.

Line 21: Improve clarity by removing semi-colon and starting a new sentence.

Introduction: A sentence listing some of the specific ecosystem services lost when wetlands are degraded would strengthen the introduction.

Introduction: What have other studies found when comparing function of natural and restored wetlands?  Context of previous studies is needed.

Introduction: The introduction would benefit from more detail and organization to provide greater clarity of ideas and context for the study.

Line 62: Do you mean that you found that the water quality index of restored wetlands was lower than that of natural wetlands?  What exactly is being compared?

Line 63: Needs clarity on what is being compared.  What does “it” refer to?

Line 65-66: Need clarification on the water quality index.  If it correlated poorly with indicators of wetland quality, why is it being used? 

Methods: Were there any significant weather events during the month that sampling took place that would affect water quality?  Because each site was sampled once over the course of that month, are there are variables that changed over time that are important to note?

Lines 133-134: Mercury sampling is not mentioned in the methods section. 

Author Response

We thank the reviewers for their useful comments, which we respond to below and make edits in the revised manuscript using Track Changes function of Word.  Our responses below are in bold text and line references in our response refer to the revised manuscript.

Reviewer 1

Comments and Suggestions for Authors

This manuscript compares the functionality of restored wetlands to that of natural wetlands in the St. Lawrence River valley in New York State.  This work is important to understand if restored wetlands can effectively reestablish the ecosystem services lost when natural wetlands are drained or degraded as a result of human activity.

The authors hypothesize that restored upland wetlands provide the same ecosystem services and biodiversity as natural wetlands in the St. Lawrence River valley.

The authors compared ten water quality measures from 17 natural and 45 restored wetlands to determine if wetland restoration results in comparable physiochemical conditions to those found in natural wetlands.  Natural wetlands were found to be more acidic and have higher fecal coliforms.  Restored wetlands had higher conductivity and related ions.  The other water quality parameters measured showed no significant difference, leading the authors to conclude that the restored wetlands function in a similar way to the natural wetlands in the region.

Specific Comments:

  1. Title: It feels like there is a disconnect between the title and the paper. The conclusion statement used as the title seems to overstate the main conclusion in the paper.  The conclusion section states that water quality differences are minor, while the title suggests large differences in wetlands that have similar functions.

The water quality does differ between the two types of wetlands.  However, these differences in some water quality parameters, which attribute to the restorative activities and nature of natural wetlands, does not have an impact on their function, as determined by biotic surveys of these wetlands in this landscape (references 6, 8, & 9)

  1. Line 21: Improve clarity by removing semi-colon and starting a new sentence.

We suspect the reviewer meant line 41 (of the revised manuscript).  We have removed the semi-colon as suggested.

  1. Introduction: A sentence listing some of the specific ecosystem services lost when wetlands are degraded would strengthen the introduction.

This has been included as suggested. A sentence describing these ecosystem values has been added (lines 39-42 of the revised manuscript).

  1. Introduction: What have other studies found when comparing function of natural and restored wetlands? Context of previous studies is needed.

We have added a brief discussion for context of other studies that have examined the ecosystem services and environmental quality of wetland restorations in comparison to natural wetlands in agricultural landscapes (lines 50-55). We have also added a little more detail on our studies (lines 63-67 of the revised manuscript)

  1. Introduction: The introduction would benefit from more detail and organization to provide greater clarity of ideas and context for the study.

Our response to comments 3 & 4 provides more context from the related studies and we believe now provides greater structure to the Introduction.

  1. Line 62: Do you mean that you found that the water quality index of restored wetlands was lower than that of natural wetlands? What exactly is being compared?

The water quality index developed by Chow-Fraser (reference #11) was used to compare natural and restored wetlandsWe have added details on what water quality parameters were used in the index, and why those are selected (lines 71-85 of the revised manuscript).

  1. Line 63: Needs clarity on what is being compared. What does “it” refer to?

We have rewritten the sentence to clarify what ‘it’ refers to (see line 80-82).

  1. Line 65-66: Need clarification on the water quality index. If it correlated poorly with indicators of wetland quality, why is it being used?

We used it because it was, per the literature, sensitive to anthropogenic disturbance and a valid indicator of wetland quality.  We found from our work that it wasn’t – this is discussed in reference [9]. We have added further clarifying discussion, see lines 82-85.

  1. Methods: Were there any significant weather events during the month that sampling took place that would affect water quality? Because each site was sampled once over the course of that month, are there are variables that changed over time that are important to note?

We were not aware of any significant weather events during the month that sampling took place that would affect water quality except thunderstorms. However, as mentioned in the Methods, sampling avoided the days following a heavy rainfall, such as a thunderstorm (lines 135-137), in order to avoid any potential effect on water quality.

  1. Lines 133-134: Mercury sampling is not mentioned in the methods section.

The mercury was sampled in a subset of these wetlands in a separate study (reference 9); as described on lines 177-180 of the revised manuscript).

Reviewer 2 Report

The manuscript provides valuable results in a topic which has an outstanding relevance for environmental management in the future – large-scale wetland restoration efforts will be needed for flood protection, water quality regulation and other climate adaptation issues, in several parts of the world. In my opinion, the quality of the analysis and its presentation could be improved according to the following:

- The introduction have to be reworked a bit – at first, the protection of biodiversity and ecosystem services is mentioned as motivation and even in the key question. But we have to remember that the water quality and its indicators are not ecosystem services themselves. The improvement in water quality, which can be measured with such metrics, might be regarded as such, but we rather have to handle them as indicators of ecosystem condition, which is on another level of assesment in the cascade system. (Please see the literature background of this, this ditinction has a high significance from an environmental management point of view. This  is especially the case with biodiversity, which can be assigned rather to ecosystem condition/helath, instead of services. This do not have direct connection with the analysis itself, but I would suggest to avoid the terminology of ecosystem services, in such an analysis.

- Methodology: the analysis is based on a measurement campaign in one single season, while there might be seasonal differences within one water body. Therefore, the differences between the investigated sample points/areas could also be affected. At least some explanation (in some sentences) would be needed to clarify, why were these measurements enough for deriving the consequences.

- I recommend to change the style of the scale bar in order to have more divisions, not only one.

Author Response

Our responses below are in bold text and line references in our response refer to the revised manuscript.

Reviewer Two

The manuscript provides valuable results in a topic which has an outstanding relevance for environmental management in the future – large-scale wetland restoration efforts will be needed for flood protection, water quality regulation and other climate adaptation issues, in several parts of the world. In my opinion, the quality of the analysis and its presentation could be improved according to the following:

  1. The introduction has to be reworked a bit – at first, the protection of biodiversity and ecosystem services is mentioned as motivation and even in the key question. But we have to remember that the water quality and its indicators are not ecosystem services themselves. The improvement in water quality, which can be measured with such metrics, might be regarded as such, but we rather have to handle them as indicators of ecosystem condition, which is on another level of assessment in the cascade system. (Please see the literature background of this, this distinction has a high significance from an environmental management point of view. This is especially the case with biodiversity, which can be assigned rather to ecosystem condition/health, instead of services. This do not have direct connection with the analysis itself, but I would suggest to avoid the terminology of ecosystem services, in such an analysis.

As suggested by Reviewer 1, the Introduction has been developed to provide greater context for the study.

  1. Methodology: the analysis is based on a measurement campaign in one single season, while there might be seasonal differences within one water body. Therefore, the differences between the investigated sample points/areas could also be affected. At least some explanation (in some sentences) would be needed to clarify, why were these measurements enough for deriving the consequences.

Although this analysis is based on a measurement campaign in a single season, we acknowledge that there might be seasonal differences within one waterbody.  Repeated measurements of water quality parameters over a summer season in a smaller subset of these wetlands showed that water quality parameters were consistent between sampling dates over 5 months [13]. Thus, we believe that single visit to this larger set of wetlands over a short duration (1 month)  was adequate for assessing differences. This is stated on line 170-176.

  1. I recommend to change the style of the scale bar in order to have more divisions, not only one.

We believe the scale bar is adequate for the purposes of this figure.